# Comparative Study of Potential Habitats for *Simulium qinghaiense* (Diptera: Simuliidae) in the Huangshui River Basin, Qinghai–Tibet Plateau: An Analysis Using Four Ecological Niche Models and Optimized Approaches

**DOI:** 10.3390/insects15020081

**Published:** 2024-01-23

**Authors:** Yunxiang Liu, Chuanji Li, Hainan Shao

**Affiliations:** 1State Key Laboratory of Plateau Ecology and Agriculture, Academy of Agricultural and Forestry Sciences, Qinghai University, Xining 810016, China; 17791394452@163.com (Y.L.); lichuanji0103@163.com (C.L.); 2Provincial Key Laboratory of Agricultural Integrated Pest Management in Qinghai, Academy of Agricultural and Forestry Sciences, Qinghai University, Xining 810016, China

**Keywords:** Qinghai-Tibet Plateau, indicator insects for environmental assessment, ecological niche model, climatic factor, potential area

## Abstract

**Simple Summary:**

*Simulium qinghaiense*, endemic to the Huangshui River, serves as a critical environmental indicator for assessing the ecological health of both the river and its surrounding landscapes. Notably, the female adults of *S. qinghaiense* could directly cause severe economic losses for farmers. Despite its ecological importance, there is currently a gap in research regarding the potential areas of this species within the region. This study marks the first comprehensive integration of the MaxEnt model (with parameter optimization), GARP, BIOCLIM, and DOMAIN models. Utilizing actual survey data on the geographical distribution of *S. qinghaiense* in the Huangshui River Basin, coupled with bioclimatic and altitudinal variables, we conduct a comparative prediction of the potential areas for this species. All four models exhibit excellent predictive accuracy, surpassing random models, with MaxEnt showcasing superior performance. The primary concentration of suitable areas is observed in the central and southern regions of the Huangshui River Basin. The MaxEnt model is then employed to simulate predictions of distribution and changes across different periods, revealing that the Qilian Mountains may provide a potentially favorable refuge for this species during the ice age. In conclusion, the findings of this study offer a scientific foundation for ecosystem conservation in the Huangshui River Basin, as well as monitoring and early warning for threshold densities of *S. qinghaiense*.

**Abstract:**

The Huangshui River, a vital tributary in the upper reaches of the Yellow River within the eastern Qinghai–Tibet Plateau, is home to the endemic black fly species *S. qinghaiense*. In this study, we conducted a systematic survey of the distribution of the species in the Huangshui River basin, revealing its predominant presence along the river’s main stem. Based on four ecological niche models—MaxEnt with parameter optimization; GARP; BIOCLIM; and DOMAIN—we conduct a comparative analysis; evaluating the accuracy of AUC and Kappa values. Our findings indicate that optimizing parameters significantly improves the MaxEnt model’s predictive accuracy by reducing complexity and overfitting. Furthermore, all four models exhibit higher accuracy compared to a random model, with MaxEnt demonstrating the highest AUC and Kappa values (0.9756 and 0.8118, respectively), showcasing significant superiority over the other models (*p* < 0.05). Evaluation of predictions from the four models elucidates that potential areas of *S. qinghaiense* in the Huangshui River basin are primarily concentrated in the central and southern areas, with precipitation exerting a predominant influence. Building upon these results, we utilized the MaxEnt model to forecast changes in suitable areas and distribution centers during the Last Interglacial (LIG), Mid-Holocene (MH), and future periods under three climate scenarios. The results indicate significantly smaller suitable areas during LIG and MH compared to the present, with the center of distribution shifting southeastward from the Qilian Mountains to the central part of the basin. In the future, suitable areas under different climate scenarios are expected to contract, with the center of distribution shifting southeastward. These findings provide important theoretical references for monitoring, early warning, and control measures for *S. qinghaiense* in the region, contributing to ecological health assessment.

## 1. Introduction

Global climate change has exerted a profound influence on biodiversity and the patterns of biological distribution [1]. Many species, subject to the pressures of natural selection, have evolved adaptive strategies to cope with environmental changes [2], thereby emphasizing the pronounced complexity and dynamism of both biodiversity and the global environment [3]. Through adjustments to the adaptability and distribution patterns of species within ecosystems, biodiversity can continually respond to the shifts brought about by global climate change [1]. Freshwater ecosystems play a pivotal role in upholding the stability of the natural world [4,5]. Climate change has significantly impacted freshwater ecosystems [6], with alterations in precipitation patterns and temperature reshaping the structure and functionality of these ecosystems in various regions. Consequently, this has implications for the biodiversity and distribution patterns of species within these environments [7]. Understanding how freshwater ecosystems and their biodiversity respond to climate change and the subsequent impact on global biodiversity conservation holds particular significance. Insects, serving as a microcosm of Earth’s biodiversity, possess a rich history intricately linked with paleoecology, particularly during periods of climatic upheaval such as the ice ages. These events have not only molded the current global distribution patterns of species but have also profoundly shaped the evolution and distribution of insects [8]. The expansion and contraction of glaciers have given rise to biological refuges, guiding species migration and propelling distinct patterns of biodiversity [9]. A thorough comprehension of these historical events is indispensable for uncovering existing patterns of insect dispersal and diversity, thus laying the groundwork for predicting the ecological impacts of future environmental changes [8,9,10]. 

The Huangshui River, situated in the eastern part of the Qinghai–Tibet Plateau, stands as a critical freshwater ecosystem in the upper reaches of the Yellow River, supporting around 60% of Qinghai Province’s population [11]. Preserving the ecological environment, managing water resources, and fostering high-quality development in eco-livestock represent major strategic objectives for the region’s future [11]. Extensive research on the assessment of environmental quality utilizes the abundance of environmental indicator insects in freshwater ecosystems. The quantity and quality (i.e., environmentally sensitive vs. tolerant) of these insects not only vividly reflect the health of ecosystems but also aid researchers in predicting and addressing a series of impacts resulting from global climate change, providing vital information for addressing this global challenge [12,13,14]. Black flies (Diptera: Nematocera: Simuliidae) are distributed in the Huangshui River basin and have been identified as environmental indicator species crucial for evaluating the ecological health of rivers and surrounding landscapes [15]. They play a pivotal role in river ecosystems [15]. This group undergoes complete metamorphosis, with larvae and pupae residing in water, feeding on organic particles in the water flow, and attaching to various substrates. Their high population densities make them a key element in the energy transfer of river food chains, serving as a food source for many vertebrates and invertebrates [16]. After eclosion, these insects leave the water and adapt to terrestrial life [17,18]. Globally, the black flies are diverse, with over 2200 known species [19] and distributed worldwide except Antarctica [20,21,22,23,24,25]. In China, there are 209 known species of Simuliidae, primarily distributed in the Northeast, South China, North China, Central China, Southwest, Mongolia-Xinjiang, and Qinghai–Tibet, indicating a relatively broad distribution [26]. In Qinghai Province, there are approximately 20 species of Simuliidae, with *S. qinghaiense* being the predominant species and endemic to the Huangshui River basin [27]. 

European researchers have investigated the distribution of black flies (Simuliidae) in Mediterranean rivers, noting their high susceptibility to river pollution and their distribution along a gradient of pollution [15,28]. In China, research on Simuliidae is relatively limited, focusing primarily on describing the taxonomic characteristics of new species, conducting geographical distribution surveys, and investigating their life history and biological characteristics [26,29,30]. Moreover, black flies have gained notoriety due to the hematophagous tendencies exhibited by adult females across a multitude of species [31]. Beyond merely posing a nuisance to humans as well as domestic and wild animals, these flies are recognized as vectors for diseases, including bovine onchocerciasis and vesicular stomatitis virus in livestock [32,33]. In tropical regions, anthropophilic species are involved in the transmission of mansonelliasis, a filarial infection, as well as onchocerciasis, colloquially known as ‘river blindness’—ranking as the world’s second-leading infectious cause of blindness [31,32,33]. Although no studies have documented the transmission of zoonotic diseases of *S. qinghaiense* to date in China, great attention should be paid to these black flies as potential vectors.

In recent years, ecological niche models (ENMs) and species distribution models (SDMs) have been widely applied to predict species responses to climate change, analyze species distribution, and understand variations therein [34]. These models significantly enhance our capacity to comprehend patterns of species distribution, making them critical in studying the crucial implications of climate change on biodiversity. Consequently, they have emerged as focal points in research across fields such as ecology, evolutionary biology, and biogeography [35]. While there are plenty of commonly used ENMs, each capable of independently forecasting potentially suitable areas, it is important to note that each model exhibits specific biases or preferences [36]. The application of these models is instrumental in devising effective conservation strategies and management approaches, thereby offering a more nuanced understanding of the intricate interplay between biodiversity and climate change [37]. Currently, research utilizing ENMs to predict the potential distribution of environmental indicator insects is widespread [38,39]. However, a notable research gap exists concerning the potential areas for environmental indicator insects in the Huangshui River basin. Existing studies are confined to exploring the community structure of benthic insects [40]. 

Given that projections by alternative models can deliver variable results [36], this study embraced the concept of an ensemble prediction system incorporating field survey data. It employs a comprehensive approach by integrating the forecasting of four models—MaxEnt (optimized using the Kuenm package); GARP; BIOCLIM; and DOMAIN—to assess the potentially suitable areas for the endemic species *S. qinghaiense* in the Huangshui River basin. A thorough comparative analysis is conducted to scrutinize the predictions derived from each of these models. Special care is taken to minimize the potential impact of false negatives or positives that might arise from the empirical selection of a single model. This study also identifies that the limitations of one model could be complemented by another, thereby enhancing the scientific robustness of the predictions. The primary objective is to provide a more accurate forecast of the potential areas for *S. qinghaiense* in the region, coupled with a quantification of the risk level associated with this species in the basin. Moreover, employing the MaxEnt model, this research forecasts and analyzes the potential shifts in suitable areas and distribution center migrations for *S. qinghaiense* under historical climatic conditions (LIG and MH) and future climate scenarios (2041–2060 and 2081–2100). This analysis aims to shed light on the impact of paleoecology and historical climates on the existing distribution pattern of the species. By clarifying the potential areas of *S. qinghaiense* in the Huangshui River basin, this study provides crucial theoretical insights. It also contributes to the development of monitoring and control strategies for black flies in the region, offering a scientific foundation for the prevention of the potential vectors of zoonotic diseases.

## 2. Materials and Methods

### 2.1. Sample Collection and Acquisition of Geographic Distribution Data

The Huangshui River spans a length of 374 km with a basin area of approximately 33,000 km^2^ [41,42]. From May to September during the years 2021 to 2023, this study conducts field surveys and collections of *S. qinghaiense* along the Huangshui River basin (covering approximately 300 km along the river) and surrounding areas (Qinghai Province: Haibei (HB) Tibetan Autonomous Prefecture (Gangcha county (GC), Haiyan county (HYA), Menyuan county (MY)), sections of the Qilian Mountains (QLM: Qilian county (QL), Tianjun county (TJ)), Datong county (DT), Xining city (XN), Huangyuan county (HY), Huzhu county (HZ), Huangzhong county (HZA), Pingan district (PA), Ledu district (LD), Minhe county (MH) and Gansu Province: Honggu district (HG), Yongjing county (YJ), Yongdeng county (YD), Tianzhu county (TZ)) (Figure 1; Appendix A). A GPS (Garmin ETREX221x, Garmin, Shanghai, China) device is used to record the latitude and longitude information of the areas where *S. qinghaiense* is identified within the basin, thereby acquiring the current geographic distribution data (Appendix A). To address potential issues related to data overfitting, ENMTools is employed to filter the distribution data, resulting in a total of 30 *S. qinghaiense* data points (Appendix A; Figure 1). The data are then input into Excel and saved in .csv format. Elevation data are sourced from the Geographic Spatial Data Cloud (http://www.gscloud.cn, accessed on 2 March 2023), and map information is retrieved from the National Basic Geographic Information Database (https://www.ngcc.cn/ngcc/, accessed on 2 March 2023). 

### 2.2. Software for ENMs and Related Tools

The four ENM software utilized in this study are MaxEnt (v3.4.4) (https://biodiversityinformatics.amnh.org/open_source/maxent/, accessed on 5 March 2023), Desktop GARP (v1.1.6) (http://www.nhm.ku.edu/desktopgarp/, accessed on 6 March 2023), and the BIOCLIM and DOMAIN modules within DIVA-GIS (v7.5) (http://www.diva-gis.org, accessed on 8 March 2023). ArcGIS (v10.4) (https://www.esri.com/en-us/home, accessed on 5 March 2023) and SPSS are employed for data processing.

### 2.3. Selection and Filtering of Environmental Variables

Data were acquired for 19 bioclimatic variables and one altitude variable at 30 s precision for five periods (Last Interglacial (LIG: 130,000–115,000 years ago), Mid-Holocen (MH: 6000 years ago), current, 2041–2060, 2081–2100) from the World Climate Database (https://www.worldclim.org, accessed on 10 April 2023) (Appendix A). However, the Last Glacial Maximum (LGM) data has a resolution of only 2.5 min, which is considered inadequate. Simulations for the LGM period are excluded for reliability, ensuring that all climate variable data for all periods maintain a 30 s precision. For LIG and MH, the Community Climate System Model Version 4 (CCSM4) atmospheric circulation model is chosen. For future climate variables, this study adapts the Shared Socioeconomic Pathways (SSP) from the Sixth Coupled Model Intercomparison Project (CMIP6). The simulation is conducted using the second-generation climate system model (BCC-CSM2-MR), an enhanced and upgraded version of the first-generation model (BCC-CSM1-1) developed by the Beijing Climate Center [43,44]. BCC-CSM2-MR exhibited significantly improved resolution both horizontally and vertically, making it more suitable for handling high-resolution simulation results and simulating smaller-scale physical processes. Additionally, it provides more accurate simulation results for meteorological and climate variables, with the added capability to comprehensively simulate internal interactions and feedback within the climate system [45]. Building upon CMIP5, CMIP6 incorporates shared socioeconomic and land-use scenarios. Compared to CMIP5, it provides more comprehensive information on CO_2_ concentrations and radiative forcing, thereby enhancing the accuracy and scientific validity of future climate change predictions [46]. In this study, three scenarios from CMIP6 are utilized, representing low, medium, and high forcing scenarios: SSP126, SSP370, and SSP585. 

The inclusion of an excessive number of environmental variables in ENMs may give rise to overfitting. This is because an abundance of variables increases the model’s complexity, leading it to overfit the existing data and, consequently, perform less effectively when generalized to new data, thereby reducing the model’s accuracy [46]. The potential issues of autocorrelation arising from environmental variables are negligible because autocorrelation has low effects on correlation analysis. Therefore, in this study, the Jackknife method within the MaxEnt model is initially employed to assess the contribution of all climate variables. Subsequently, a Pearson correlation analysis is conducted on the environmental variables using SPSS (Appendix A), and variables with |r| > 0.8 and low contribution rates are excluded [47]. Ultimately, eight key environmental variables are selected for subsequent model analyses (Appendix A). 

### 2.4. Optimizing the MaxEnt Model

The MaxEnt model has found widespread application across various fields. Optimizing the model by selecting the best parameters is crucial for improving prediction accuracy, reducing overfitting, and enhancing model reliability [48]. Among the most important parameters are feature combination (FC) and regularization multiplier (RM). FC provides five selectable options: Linear features (L), Quadratic features (Q), Product features (P), Threshold features (T), and Hinge features (H). The RM parameter is typically set below 4 [49]. In this study, the Kuenm package in R is utilized (https://github.com/marlonecobos/kuenm, accessed on 5 March 2023) [50], incorporating 29 combinations from FC, ranging from 0.1 to 4 at intervals of 0.1. This resulted in a total of 40 RM settings. A thorough screening involves a total of 1160 combinations, and the optimal parameters are chosen based on three criteria: statistical significance, omission rate ≤5%, and delta AICc values ≤ 2 [50].

### 2.5. Constructing the Four Models: MaxEnt, GARP, BIOCLIM, and DOMAIN

Based on the selected environmental variables and distribution data, the construction of the four models are outlined below:

MaxEnt Model: The distribution data in .csv format and environmental variables in .asc format are imported into the software. The optimal combination of FC and RM is selected. Seventy-five percent of the data are assigned as the training dataset, with the remaining 25% as the testing dataset. The Jackknife test is employed to assess the importance of environmental variables and calculate the response curves of dominant environmental factors. This process is repeated ten times.

GARP Model: The .asc format environmental variables are converted to .raw format, generating an environmental variable dataset in .dxl format. This dataset, along with the distribution data, are imported into GARP. Seventy-five percent of the data are designated as the training dataset and 25% as the testing dataset. The process is repeated 20 times.

BIOCLIM and DOMAIN Models: In ArcGIS, the distribution data in .csv format is converted to .shp format and then imported into DIVA-GIS. The .asc format environmental variables are transformed into .grd format within DIVA-GIS, and a stacked environmental variable dataset is created. In Modeling-Bioclim/Domain, the stacked environmental dataset is loaded, and both models are run sequentially.

### 2.6. Accuracy Evaluation of the Four Models

The Receiver Operating Characteristic (ROC) Curve is employed to depict the predictive capability of the models, and the Area Under the Curve (AUC) of the ROC curve is commonly used as a measure of model accuracy. The AUC value ranges from 0 to 1, with a higher value indicating more precise model predictions. Kappa value is another metric used to assess the classification performance of the models, mainly focusing on the consistency between predicted and observed results. The Kappa value ranges from −1 to 1, where negative values suggest that the model predictions are inferior to random, 0 indicates consistency between the model and random predictions, and positive values imply that the model predictions outperform random classification. A value closer to 1 signifies higher prediction accuracy. While AUC quantifies the model’s performance across various classification thresholds independently of specific thresholds, Kappa values often exhibit bias due to the influence of species distribution and diagnostic thresholds. Therefore, in this study, AUC is considered the primary evaluation metric, with Kappa serving as a supplementary metric for the comparative assessment of the accuracy of the four models [51,52]. The predictive results obtained from the four models are compiled into a stacked dataset, which is then imported into DIVA-GIS along with the distribution data. The Show ROC/Kappa module is used to output both evaluation metrics [53,54]. 

### 2.7. Prediction of Potential Distribution and Identification of Suitable Habitat

In the results obtained from the MaxEnt model, the average of 10 runs is selected as the final outcome. For the GARP model, the 20 results are overlaid in ArcGIS and averaged to obtain continuous probability values ranging from 0 to 1, which are considered the definitive result. The outcome with the highest AUC value is chosen as the final result for both the BIOCLIM and DOMAIN models. The result files from the four models are converted into raster data using ArcGIS for visualization. The Natural Breaks method, a commonly employed data classification technique particularly suitable for handling continuous numerical data with distinct clustering characteristics, is utilized. This method aims to determine optimal division points for the data by maximizing the similarity of data values within each category while minimizing the similarity between categories. Given the diverse algorithms used in different models, there are currently no standardized methods for classification. Therefore, this study employs Natural Breaks to reclassify the prediction results. The potential distribution of *S. qinghaiense* in the Huangshui River Basin is categorized into four classes: unsuitable area, low suitable area, medium suitable area, and high suitable area [55,56]. Following reclassification, a floating-point field is added to the attribute table of the result file, and different suitable areas are calculated using pixel size and VB script in the field calculator. 

### 2.8. Changes in Suitable Habitats and Centroid Migration in Different Periods

The MaxEnt model is widely recognized for its proficiency in predicting past and future periods, as well as assessing alterations in suitable areas and centroid migration. To facilitate these analyses, we employed an open-source toolkit known as SDMtoolbox (Species Distribution Model Toolbox) (http://www.sdmtoolbox.org, accessed on 12 April 2023). Specifically designed for handling spatial data in biogeography and landscape mobility studies, SDMtoolbox proves invaluable for creating and analyzing distribution models tailored to specific biological populations [57]. In this study, we employed the MaxEnt model to project predictions from the current period to different timeframes and climate scenarios. The SDMtoolbox was integrated into ArcGIS for data visualization [46,58,59]. 

## 3. Results

### 3.1. Optimization Results of the MaxEnt Model

The execution of the Kuenm package generates a total of 1,160 candidate models (Figure 2A). Assessing these models based on three criteria—statistical significance; omission rate; and AIC—reveals only one candidate model that satisfied all selection criteria. This specific model incorporates Linear features (L) and Quadratic features (Q), with an RM set at 0.6. Under these parameters, the model demonstrates no omission rate and a Delta AICc of 0 (Table 1). In contrast, the default parameters of the MaxEnt model, featuring the combination of LQHPT variables and an RM of 1, result in an omission rate of 0.375 and a Delta AICc of 64.74020, both higher than the optimized parameters. Running the MaxEnt model with the optimized parameters produces an average AUC value of 0.976 (Figure 2B). The optimization of parameters effectively mitigates the complexity and overfitting associated with the MaxEnt model, thereby significantly improving prediction accuracy. 

### 3.2. Evaluation of Model Accuracy

A one-way analysis of variance (ANOVA) is conducted to compare the AUC and Kappa values among the four models (MaxEnt, GARP, BIOCLIM, and DOMAIN) (Table 2; Figure 3). The results indicate that both *p* (AUC) = 0.199 and *p* (Kappa) = 0.995 exhibit significance greater than 0.05, meeting the assumption of homogeneity of variance. Among the four models, the MaxEnt model exhibits the highest mean AUC and Kappa values, with 0.9756 and 0.8118, respectively. In contrast, the DOMAIN model demonstrates the lowest values, with 0.8909 for AUC and 0.6780 for Kappa (Table 2). The standard deviations of AUC values follow the order MaxEnt < DOMAIN < BIOCLIM < GARP, while the standard deviations of Kappa values follow MaxEnt < BIOCLIM < GARP < DOMAIN (Table 2). LSD multiple comparisons reveal significant differences between the AUC and Kappa values of the MaxEnt model and the other three models (*p* < 0.05) (Figure 3A). However, there are no significant differences in AUC values among GARP, BIOCLIM, and DOMAIN models and no significant differences in Kappa values between GARP and BIOCLIM (*p* = 0.72) (Figure 3A). Furthermore, both the AUC and Kappa values of the four models exhibit a normal distribution (Appendix A; Figure 3B). The AUC values of the MaxEnt model are concentrated around 0.98, while the Kappa values of the MaxEnt, GARP, BIOCLIM, and DOMAIN models are more dispersed (Figure 3B). In summary, all four models demonstrate predictive accuracy surpassing that of random models, with the MaxEnt model exhibiting superior performance, followed by GARP and BIOCLIM, while the DOMAIN model showed slightly inferior performance.

### 3.3. Analysis of Dominant Environmental Factors

Through an assessment of the contribution rates and permutation importance of environmental variables obtained by the MaxEnt model, it is identified that the cumulative contribution of four key variables—precipitation of the warmest quarter; precipitation of the driest month; altitude; and annual precipitation—reaches 91.8%; with an overall permutation importance of 88.7% (Appendix A). Taking into account the results from the Jackknife test (Appendix A), it is observed that, when using individual variables, the most significant impact on normalized training gain comes from annual precipitation and precipitation of the warmest quarter, followed by altitude and annual precipitation. The absence of annual precipitation and precipitation in the warmest quarter leads to a notable reduction in normalized training gain, indicating the provision of unique information not covered by other environmental variables (Appendix A). Consequently, the key environmental factors influencing *S. qinghaiense* in the Huangshui River basin are determined to be precipitation of the warmest quarter, precipitation of the driest month, altitude, and annual precipitation (Appendix A). When these factors fall within the favorable range, exceeding a probability of 0.5, it enhances the suitability for the survival of *S. qinghaiense* in the Huangshui River basin. Analyzing the response curves of these dominant environmental factors (Figure 4): Precipitation in the warmest quarter is conducive to survival in the range of 192–280 mm, with the optimum value at 236 mm; Altitude in the range of 1780–3192 m is favorable for survival, with the optimum value at 2480 m; and Annual precipitation in the range of 348–475 mm is favorable for survival, with the optimum value at 412 mm. Precipitation in the driest month is beneficial for survival in the range of −0.9–0.774 mm. Given that negative precipitation values do not exist in reality, considering the model’s limitations, the suitable range for precipitation in the driest month is between 0 and 0.774 mm, with the optimum value at 0 mm (Figure 4).

### 3.4. Predicted Potential Distribution of S. qinghaiense by Four Models

The potential areas for *S. qinghaiense*, as predicted by the four models in the Huangshui River basin, are summarized below (Table 3; Figure 5 and Figure 6). MaxEnt Model: The prediction indicates that high suitable areas are mainly distributed in the central (XN, HZ, DT, and HZA) and southern (HZA, PA, and LD) regions of the Huangshui River basin, while unsuitable areas are concentrated in the northern (HB and QLM) and western (QLM and HYA) regions (Figure 1 and Figure 5). The proportions of unsuitable, low suitable, medium suitable, and high suitable areas are 32.94%, 26.32%, 20.83%, and 19.90%, respectively (Figure 6). GARP Model: The prediction results show that the unsuitable area and the high suitable area are the largest among the four models, covering 15,990.95 km^2^ and 10,327.08 km^2^, respectively (Table 3; Figure 5). High suitable areas are mainly distributed in the central (XN, HZ, DT, and HZA), southern (HZA, PA, and LD), and eastern (MH, YJ, YD, and HG) parts of the Huangshui River basin, while unsuitable areas are concentrated in the northern (HB and QLM) and western (HYA and QLM) regions. The proportions of unsuitable, low suitable, medium suitable, and high suitable areas are 48.01%, 13.53%, 7.44%, and 31.01%, respectively (Figure 6). BIOCLIM Model: The predicted high suitable areas are mainly concentrated in the local regions of MY, HZ, HZA, and HY, while unsuitable areas are prevalent in most parts (QLM, XN, DT, PA, LD, TZ, MH, YJ, YD, and HG) of the Huangshui River basin (Figure 1 and Figure 5). Notably, the high suitable area is the smallest among the four models, covering only 1896.55 km^2^ (Figure 5). 

The proportions of unsuitable, low suitable, medium suitable, and high suitable areas are 40.75%, 34.88%, 18.68%, and 5.69%, respectively (Figure 6). DOMAIN Model: The prediction results show that the medium suitable area is the largest among the four models, reaching 13,778.23 km^2^ (Table 3; Figure 5). The predicted suitable areas cover various parts of the Huangshui River basin. The proportions of unsuitable, low suitable, medium suitable, and high suitable areas are 18.09%, 16.50%, 41.37%, and 24.05%, respectively (Figure 6). Based on these results, the MaxEnt model provides a more balanced prediction with smaller differences in the areas of the four types of suitable areas. The GARP model mainly focuses on high suitable and unsuitable areas, with a relatively smaller square measure for medium suitable and low suitable areas (20.98%). The BIOCLIM model predicts the smallest suitable range, with more dispersed results and a lower proportion of high suitable areas. The DOMAIN model predicts the widest suitable range, with medium suitable and high suitable areas accounting for 65.42% (Figure 6). The predictions of high suitable areas using MaxEnt, GARP, and DOMAIN models are similar, and MaxEnt and GARP models exhibit similar predictions for unsuitable areas.

### 3.5. Predicted Suitable Areas and Their Changes over Different Periods and Climate Scenarios

During the LIG period, the predicted suitable areas for *S. qinghaiense* in the Huangshui River basin are primarily situated in the Qilian Mountains (QLM). The unsuitable areas cover 24,245.1 km^2^, constituting 72.8% of the total area, marking a 54.74% increase compared to the current conditions (Table 4; Figure 7). The combined square measure of low suitable, medium suitable, and high suitable areas is only 9059.79 km^2^, with a significant decrease observed in the high suitable area of 694.64% (Table 4). Notably, there is an expansion of the suitable area near the eastern edge of the Qilian Mountains (MY), with an increase of approximately 315.15 km^2^ (Table 4; Figure 7). Despite this local expansion, the overall suitable areas exhibit a substantial contraction of about 23,011.56 km^2^ compared to the current conditions (Table 4). Transitioning to the MH period, the predicted suitable areas for the species are mainly concentrated in the current regions of DT, XN, and HZA (Figure 1 and Figure 7). In comparison to the LIG period, the unsuitable area decreased, covering 57.31% of the total area, representing a 42.51% increase compared to current conditions (Table 4). Concurrently, the total area of low suitable, medium suitable, and high suitable increases to 14,219.49 km^2^ (Table 4). The suitable area near the eastern edge (MY, DT, and HYA) of the Qilian Mountains expands by a total of 187.7 km^2^. Although a considerable contraction still occurred compared to the current situation, the degree of contraction significantly decreased compared to the LIG period, reaching 15,902.86 km^2^ (Table 4).

During the period of 2041–2060, under three climate scenarios (SSP126, SSP370, and SSP585), the most suitable areas for the species are primarily concentrated in the central (XN, HZ, DT, and HZA) and southern (HZA, PA, and LD) regions of the basin. In comparison to current conditions, HG no longer falls within the high suitable area (Figure 1 and Figure 8A). Unsuitable areas, mainly HB and QLM, have increased in size, with increments of 24.21%, 16.73%, and 14.86% compared to the present state. In the medium suitable and low-suitable areas, there is an overall increasing trend in square measure, except for a decrease of 4.69% and 3.48% under SSP370 and SSP585, respectively (Table 4). Regardless of the climate scenario, the highly suitable area exhibits a significant reduction in square measure compared to current conditions, with the reduction sequence being SSP126 (53.55%) > SSP585 (49.62%) > SSP370 (39.22%). The respective areas are 3078.47 km^2^, 4028.49 km^2^, and 3338.86 km^2^ (Table 4). It is noteworthy that under the SSP370 and SSP585 scenarios in the future, the suitable areas will slightly increase compared to current conditions, with increments of 9.27 km^2^ and 454.19 km^2^, respectively (Table 4; Figure 8B). Regarding stable suitable areas, the order of the three scenarios is SSP585 > SSP370 > SSP126, and in terms of contraction areas, SSP126 is the largest, followed by SSP370, and SSP585 is the smallest, with specific values of 5938.47 km^2^, 2250.87 km^2^, and 1949.62 km^2^, respectively (Table 4). 

In the period from 2081 to 2100, under three climate scenarios (SSP126, SSP370 and SSP585), the suitable areas for *S. qinghaiense* in SSP126 and SSP370 remain largely consistent with those of 2041–2060 (Table 4; Figure 8A). However, under SSP585, the high-suitable area is predominantly concentrated in the central part (XN, HZ, DT, and HZA) of the basin. In comparison to SSP126 and SSP370, there is a significant reduction in area, with the most substantial decrease reaching 78.04%. The highly suitable area is diminished to only 1455.55 km^2^ (Table 4; Figure 8A). The unsuitable area maintains a similar distribution in QLM during the current period, 2041–2060, and 2081–2100. Nevertheless, the increase in the unsuitable area is significantly higher in 2081–2100, showing increments of 24.91%, 71.65%, and 102.25% compared to the current period (Table 4). Concerning the medium suitable and low suitable areas, except for a minor increase under SSP126, both SSP370 and SSP585 exhibit varying degrees of reduction. The reduction is particularly severe under SSP585 (Table 4). In 2081–2100, the suitable area for this species only expands under SSP370, primarily concentrated in some areas of QLM (TJ and MY), with a total expansion area of 624.13 km^2^ (Table 4; Figure 8B). The stable suitable area is ranked as SSP126 > SSP370 > SSP585, while the contraction area is ranked as SSP585 > SSP370 > SSP126 (Table 4). Based on these results, in comparison to current conditions, the suitable area of *S. qinghaiense* significantly decreases during LIG and partially recovers in MH (Figure 8), indicating a substantial impact of glacial activities on the suitable area of this species during ancient times. In future periods, the suitable area will continue to decrease, especially under SSP585. Meanwhile, there is a slight trend of expansion in the suitable area, particularly under SSP370 and SSP585. The prediction for 2081–2100 indicates a continuous contraction of the suitable area, with localized expansion under SSP370 (Figure 8). This suggests that climate change has a significant impact on the suitable area of this species and will continue to undergo changes in the future. 

### 3.6. Distribution Center Changes over Different Periods

The distribution center of *S. qinghaiense* in the Huangshui River Basin has undergone significant changes over various periods and under different climate scenarios. Currently, this species is located at 36.84° N, 101.89° E, in the central region (HZ) of the basin (Figure 9; Appendix A). During paleoclimatic period periods (LIG and MH), the distribution center was situated at 37.73° N, 100.71° E, and 36.88° N, 101.43° E, respectively, in the northwest region of the basin (Qilian Mountains’ foothills (QL)) and the central region of the basin (HZA) (Figure 9; Appendix A). The distribution center gradually shifted southeastward, moving from QL to HZA and further southeast to its current position (HZ) (Figure 9).

For future climate scenarios, predictions indicate different characteristics in the location and migration trends of the distribution center. According to predictions under SSP126 and SSP370, the distribution center consistently remains in HZ (Figure 9). Under SSP126 and SSP370, during the periods of 2041–2060 and 2081–2100, the distribution center shows a minor degree of migration (Figure 9; Appendix A). Under SSP585, during the period of 2041–2060, the distribution center will migrate northwestward by 7.97 km from its original position to 36.85° N, 101.83° E. By 2081–2100, the distribution center will significantly shift southeastward by 28.15 km, ultimately reaching LD (36.57° N, 102.31° E) (Figure 9; Appendix A). In summary, the distribution center of *S. qinghaiense* in the Huangshui River Basin exhibits a clear trend of unidirectional migration during the LIG period, and future predictions suggest an overall trend of slight southeastward migration. 

## 4. Discussion

This study reveals significant differences among the models, with the MaxEnt model exhibiting the highest accuracy. Further optimization of the MaxEnt model using the Kuenm package emphasized the crucial role of parameter optimization in predictive modeling. It explores the inherent importance of typical complex environmental conditions and affirms the applicability of machine learning in ecological predictive modeling. The impact of parameter selection and optimization in the MaxEnt model is highlighted, particularly in the choice of RM, effectively constraining model complexity, enhancing operational capabilities, and avoiding overfitting [58]. In this study, when the feature combination is Linear features (L) and Quadratic features (Q) with an RM of 0.6 (Table 1), the predictive performance of the model significantly improves. Some scholars underscore the importance of statistical significance, omission rate indicators, and AICc criteria on model selection. The model results are considered acceptable only when all these criteria are met [48]. In evaluating model performance, the inclusion of performance assessment metrics ensures a comprehensive measurement of prediction results from various perspectives [59]. Therefore, a comprehensive consideration of three evaluation indicators—statistical significance; omission rate; and AICc (Table 1)—is undertaken to ensure the obtained conclusions exhibit high accuracy and reliability. Furthermore, this study draws inspiration from the perspective of Muscarella et al. [60], emphasizing that candidate models, when predicting species distribution, can provide a wealth of valuable parameter sets, potentially serving as a powerful tool for model optimization. These studies further validate the accuracy of the results in this research, elucidating essential principles regarding model selection and parameter optimization, thus providing robust guidance for future ecological predictive modeling. 

The comparison of four different models reveals that the MaxEnt model stands out significantly in terms of accuracy, with AUC and Kappa values reaching 0.9756 and 0.8118, respectively (Table 2). This performance is significantly higher compared to the other three models. These findings are consistent with the studies of Elith et al. [61] and Yang et al. [47]. Furthermore, previous studies suggested that AUC and Kappa, as metrics for accuracy, typically follow a normal distribution, providing a valuable reference for assessing predictive capabilities and aligning with the results of this study [62]. One possible explanation for the DOMAIN model’s poor performance is its assumption that each species has a fixed “domain” of environmental preferences. However, actual biological distribution may be influenced by multiple environmental factors and may not always fit within a clearly defined “domain” in the selected parameter space [63]. According to the predictions of the MaxEnt model, both unsuitable areas and medium suitable or high suitable areas exhibit a relatively even distribution without a pronounced concentration trend. In contrast, the predictions of the GARP model are mainly concentrated in unsuitable and high suitable areas, with relatively smaller areas for medium or low areas. Overfitting may occur in GARP predictions, leading to an overly concentrated outcome [64]. The BIOCLIM model predicts the smallest, most suitable area among the four models, indicating a more conservative and discrete outcome. BIOCLIM defines the ideal ecological niche based on the currently observed distribution of species along environmental gradients, potentially resulting in a smaller predicted potential distribution, thus making its predictions more conservative and discrete than some modern models [65]. The DOMAIN model’s predictions exhibit the widest suitable distribution, using a method of weighted overlay for the location of species to predict the distribution in simulated environmental space, leading to a broad, high suitable area [63]. Taking various factors into account, the results of this study once again underscore the superiority of the MaxEnt model in predicting the potential distribution of organisms. This model is valuable for understanding and predicting species distribution, aiding in the formulation of targeted conservation strategies to protect biodiversity [47]. Moreover, the effective use of models does not depend on the superiority of a single model but on selecting the model most suitable for the current environment and species conditions, often through combining and comparing with other models. Therefore, future research should continue to explore more excellent models and introduce the fusion and comparison of different models [66,67]. 

Based on the analysis of dominant environmental factors, the crucial environmental factors influencing the survival of *S. qinghaiense* include precipitation in the warmest quarter, precipitation in the driest month, altitude, and annual precipitation. Precipitation significantly impacts both the parasitism and survival of Simuliidae [68]. The study results reveal that the primary factor affecting the suitable areas of *S. qinghaiense* is precipitation (Appendix A). This is likely due to the larvae of *S. qinghaiense* inhabiting swiftly-flowing water. Increased precipitation typically reduces water temperature and enhances oxygen supply, creating a more favorable habitat. Previous research has emphasized the profound influence of altitude on the distribution of Simuliidae, as both geographical and climatic factors affect the genetic differentiation and spatial distribution of this taxonomic group [31]. This aligns with our study findings, underscoring the critical role of altitude in a suitable environment for *S. qinghaiense*. Previous studies have indicated that the abundance of blackflies is higher and the young larvae exhibit faster growth rates in fast-flowing streams [69,70]. It is worth noting the influence of current velocity on *S. qinghaiense*. However, data on environmental factors is acquired from WorldClim, where data on the current velocity of the Huangshui River Basin is lacking. Further study would take all the environmental factors into consideration. The study provides clear insights into the significant impact of environmental factors on the distribution pattern of this species in the Huangshui River Basin. This offers theoretical references for future research on the ecological processes, climate change adaptation, and interactions with human activities within this taxonomic group [71,72,73]. 

Insects have a profound connection with climate, influencing both historical and current distributions [74]. The interplay of climate oscillations with geological events serves as a critical diversifying force, deeply shaping the structure of biological communities [75]. Climate fluctuations during the LIG period created conditions for population isolation in various refugee camps [76,77]. These refuges provide stable microclimates that facilitate species survival during the extreme climate changes of the Ice Age. This allows species to shift to more favorable habitats or undergo adaptive evolution [78]. The findings of this study reveal that during the LIG period, the suitable areas of *S. qinghaiense* significantly decreased, particularly in high suitable areas (Table 4; Figure 7). This reduction is likely attributed to intense climate fluctuations during the LIG period, leading to alterations in the geographical distribution and suitable environmental conditions for the species. The drastic climate changes impede the evolutionary processes of species, exerting a substantial impact on species development and potentially resulting in a significant reduction in species abundance [79,80]. The predictions of suitable areas under paleoclimate and the changing trend of distribution centers suggest that during the LIG period, the suitable areas and distribution centers of *S. qinghaiense* were primarily located in the hinterland of the Qilian Mountains (QLM: QL) (Figure 7 and Figure 9). Due to the harsh climate conditions during the ice age [81], this region likely serves as a favorable refuge for the species [76,77]. The diverse topography of the Qilian Mountains, with peaks and valleys, provided superior microclimatic conditions, effectively mitigating the impact of the ice age climate on the distribution of this insect. The predicted range shift of *S. qinghaiense* from the LGM period to the current climate conditions indicates that the warming temperatures post-ice age led to the gradual expansion of the species’ distribution pattern towards warmer and lower-altitude areas, particularly in the southern part of the Huangshui River Basin (Figure 9). 

The results of future climate predictions (2041–2060 and 2081–2100) indicate a significant reduction in the square measure of a highly suitable area (Figure 8). This reduction is attributed to the anticipated impacts of future climate change, particularly global warming, which is expected to profoundly influence the distribution patterns of *S. qinghaiense*. This aligns with the findings of Urban et al. [81]. In addition, significant alterations in local water quality caused by anthropogenic activities of humans, such as expanding human populations, intensive agricultural practices, and releases from industrial wastewater and domestic sewage, greatly affect the distribution patterns of immature stages of black flies in the water, whose habitats typically require clean, unpolluted water [14,82]. Water quality might be another important driver of the distribution patterns of *S. qinghaiense.* The predicted shift in the distribution center under future climate conditions reveals an overall southeastward migration. This shift may have far-reaching effects on the ecological niche of this taxon in the Huangshui River Basin. Alterations in species distribution could modify interactions between predators and prey, as well as competition dynamics with other species, thereby influencing the overall stability of the ecosystem [83,84]. It is crucial to note that predictions inherently carry uncertainties, and model results require on-site validation and support from field investigations [85]. Additionally, the models employed in this study did not account for species adaptability and inter-species interactions, factors that can impact the accuracy of predictions [86]. The study focuses solely on the influences of climate and altitude changes, omitting human activities as influencing factors. Future research under different climate scenarios should consider improvements in this regard [87,88]. 

Landis’s study underscores the importance of monitoring environmental indicator species at different scales, both spatial and temporal [89]. Future research should analyze data at various scales to gain a better understanding of the dynamic changes in *S. qinghaiense* within the Huangshui River Basin and their correlation with environmental factors. This focus on the ecosystem is essential for effective management and conservation measures, ensuring the sustainability of the entire ecosystem. Predictive analyses of indicator species can integrate diverse data sources such as remote sensing, land use, and biodiversity [90,91]. Therefore, using high-resolution ecological surveys, including the use of drones [92], will enable more precise predictions of Simuliidae distribution and associated ecological information in the Huangshui River Basin. Moreover, black flies play a crucial role as indicators of the transmission of various pathogens between humans, animals, and livestock, holding significant importance in the fields of ecological conservation and public health [32]. Changes in their abundance can reflect variations in the environmental quality of the basin, potentially leading to adjustments in the distribution and habitats of other species within the ecosystem, thereby affecting the overall ecological balance. The results of this study are expected to offer a scientific foundation for the prevention and control of the potential vectors of zoonotic diseases.

## 5. Conclusions

Through parameter optimization, the MaxEnt model’s complexity and overfitting are effectively reduced, leading to enhanced prediction accuracy. All four models exhibit excellent predictive accuracy, surpassing random models, with MaxEnt showcasing superior performance. The primary concentration of suitable areas of *S. qinghaiense* is observed in the central and southern regions of the Huangshui River Basin, while unsuitable areas are concentrated in the northwest, primarily influenced by precipitation. The MaxEnt model is then employed to simulate predictions of distribution and changes across different periods, revealing that the Qilian Mountains potentially provided a favorable refuge for this species during the ice age. In comparison to paleoclimate (LIG and MH), the current range of potential distribution is more extensive. This study not only contributes to a clearer understanding of the potential areas of *S. qinghaiense* in the Huangshui River basin but also sheds light on the influence of paleoecology and paleoclimate on its current distribution. 

## Figures and Tables

**Figure 1 insects-15-00081-f001:**
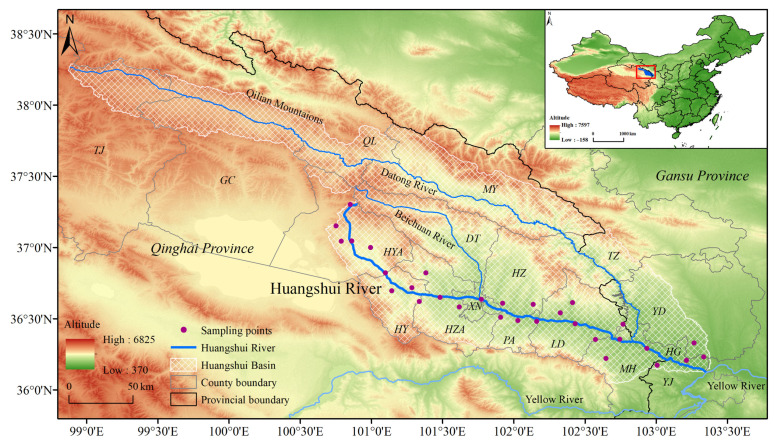
Geographic location of the Huangshui River Basin and distribution points of *S. qinghaiense*.

**Figure 2 insects-15-00081-f002:**
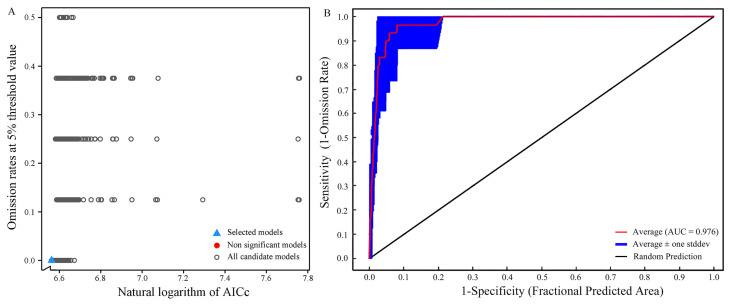
Model optimization results: (**A**) Model optimization diagram; (**B**) ROC curve of the optimized Maxent model predictions.

**Figure 3 insects-15-00081-f003:**
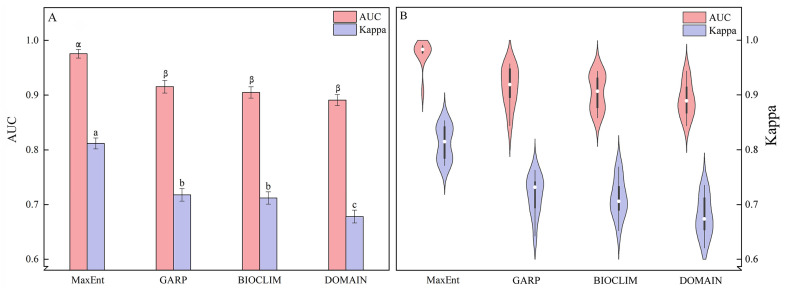
Model comparison: (**A**) Statistical comparison of AUC and Kappa values for four models; (**B**) Distribution and probability density of AUC and Kappa values for four models. Different letters indicate the significance of the different values of AUC (α, β) and Kappa (a, b, c) analyzed by the one-way analysis of variance (*p* < 0.05).

**Figure 4 insects-15-00081-f004:**
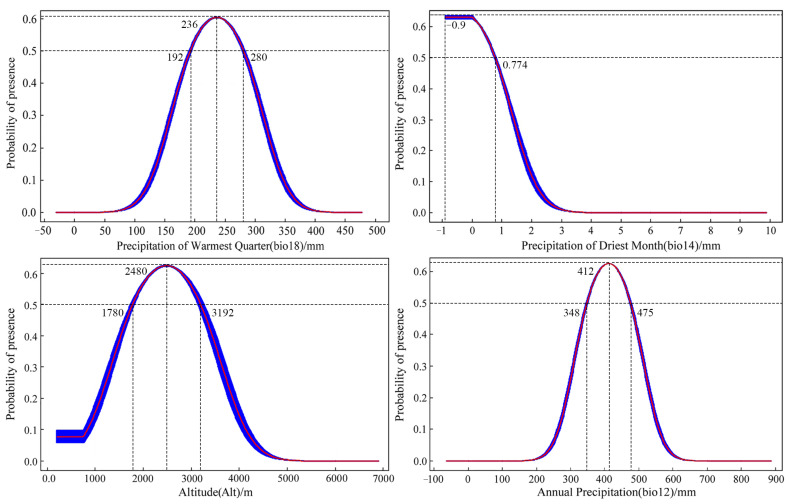
Response curves of key environmental factors.

**Figure 5 insects-15-00081-f005:**
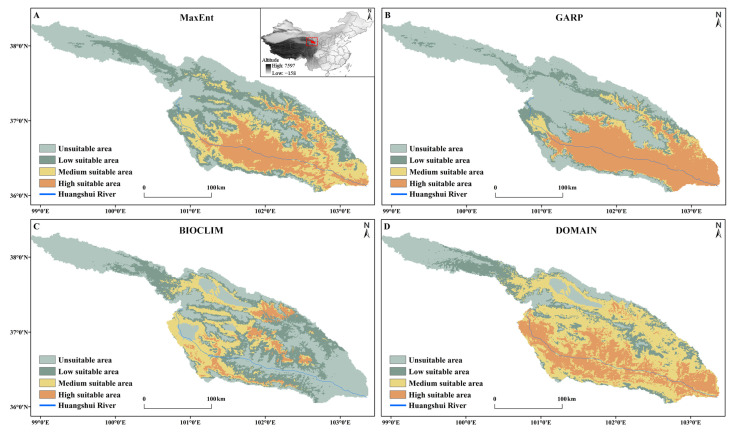
Predicted potential distribution of *S. qinghaiense* in the Huangshui River Basin using four models: (**A**) Predictive results of the MaxEnt model; (**B**) Predictive results of the GARP model; (**C**) Predictive results of the BIOCLIM model; (**D**) Predictive results of the DOMAIN model. Light green indicates unsuitable area, dark green denotes low suitable area, yellow signifies medium suitable area, and orange represents high suitable area.

**Figure 6 insects-15-00081-f006:**
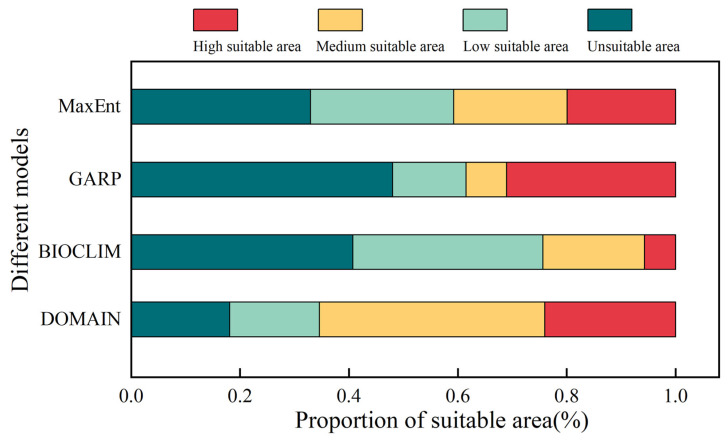
Proportional representation of suitable area under different model predictions.

**Figure 7 insects-15-00081-f007:**
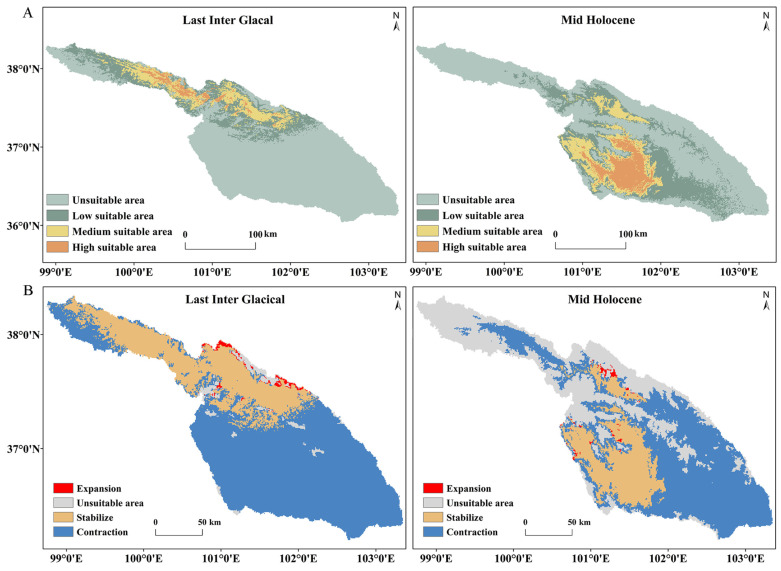
Predicted potential distribution results and their changes during LIG and MH periods: (**A**) Predictive results of potential distribution; (**B**) Changes in suitable area.

**Figure 8 insects-15-00081-f008:**
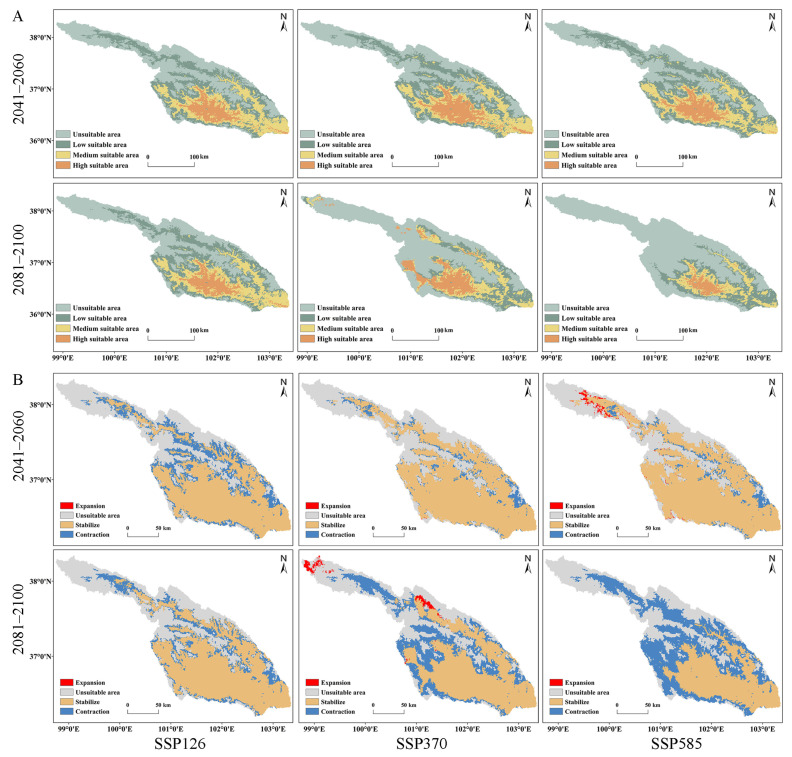
Predicted potential distribution and changes under different future periods and climate scenarios: (**A**) Predictive results of potential distribution; (**B**) Changes in suitable area.

**Figure 9 insects-15-00081-f009:**
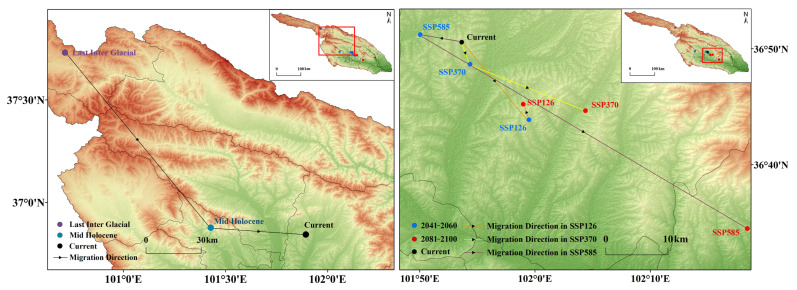
Changes in the distribution centers of *S. qinghaiense* in the Huangshui River Basin under different periods and climate scenarios.

**Table 1 insects-15-00081-t001:** Results of model optimization and selection with corresponding parameters.

Criteria	Number of Models	
All candidate models	1160	
Statistically significant models	1156	
Models meeting omission rate criteria	150	
Models meeting AICc criteria	1	
Statistically significant models meeting omission rate criteria	146	
Statistically significant models meeting AICc criteria	1	
Selected model	RM(0.6) FC(LQ)	
	Mean AUC ratio	1.9267
	Omission rate	0
	AICc	708.0172
	Delta AICc	0

**Table 2 insects-15-00081-t002:** AUC and Kappa values for four different models.

Group	MaxEnt		GARP		BIOCLIM		DOMAIN	
	AUC	Kappa	AUC	Kappa	AUC	Kappa	AUC	Kappa
1	0.9054	0.7735	0.8996	0.6877	0.9425	0.7625	0.8667	0.6426
2	0.9830	0.8314	0.8775	0.6938	0.8845	0.6933	0.8613	0.6536
3	0.9911	0.8532	0.9210	0.7324	0.8678	0.6867	0.9325	0.7352
4	0.9878	0.8325	0.9168	0.7417	0.9311	0.7236	0.9434	0.7124
5	0.9768	0.7843	0.9567	0.7359	0.9242	0.7124	0.9145	0.7243
6	0.9811	0.7893	0.8439	0.6426	0.9432	0.7686	0.8750	0.6207
7	0.9841	0.8424	0.9519	0.7512	0.8767	0.6893	0.8857	0.6550
8	0.9830	0.7982	0.9479	0.7627	0.8582	0.6521	0.8929	0.6750
9	0.9888	0.8423	0.9433	0.7310	0.9314	0.7333	0.8932	0.6883
10	0.9753	0.7711	0.8950	0.6988	0.8894	0.6993	0.8438	0.6723
Average	0.9756	0.8118	0.9154	0.7178	0.9049	0.7121	0.8909	0.6780
Standard deviation	0.0252	0.0316	0.0367	0.0364	0.0327	0.0359	0.0317	0.0371

**Table 3 insects-15-00081-t003:** Predicted suitable areas under different models.

Different Models	Area of Different Suitable Habitats (km^2^)			
	Unsuitable Area	Low Suitable Area	Medium Suitable Area	High Suitable Area
MaxEnt	10,972.2	8766.67	6938.19	6627.83
GARP	15,990.95	4507.69	2479.17	10,327.08
BIOCLIM	13,571.53	11,615.28	6221.53	1896.55
DOMAIN	6023.23	5494.42	13,778.23	8009.01

**Table 4 insects-15-00081-t004:** Suitable areas and variations under different time periods and climate scenarios.

Different Periods and Climate Scenarios	Unsuitable Area (km^2^)	Change (%)	Low Suitable Area (km^2^)	Change (%)	Medium Suitable Area (km^2^)	Change (%)	High Suitable Area (km^2^)	Change (%)	Expansion (km^2^)	Stabilize (km^2^)	Contraction (km^2^)
Current	10,972.2	-	8766.67	-	6938.19	-	6627.83	-	-	-	-
Last Inter Glacial	24,245.1	−54.74%	5362.53	63.48%	2863.19	142.32%	834.07	694.64%	315.15	8529.21	23,011.56
Mid Holocene	19,085.4	−42.51%	7984.03	9.80%	3622.27	91.54%	2613.19	153.63%	187.7	6180.24	15,902.86
SSP126 (2041–2060)	13,628.5	24.21%	9582.64	9.31%	7015.28	1.11%	3078.47	−53.55%	0	16,144.63	5938.47
SSP370 (2041–2060)	12,807.6	16.73%	9856.29	12.43%	6612.51	−4.69%	4028.49	−39.22%	9.27	19,832.23	2250.87
SSP585 (2041–2060)	12,602.8	14.86%	10,666.7	21.67%	6696.53	−3.48%	3338.86	−49.62%	454.19	20,133.48	1949.62
SSP126 (2081–2100)	13,704.9	24.91%	9475.69	8.09%	7148.61	3.03%	2975.69	−55.10%	0	17,294.01	4789.09
SSP370 (2081–2100)	18,834	71.65%	5488.19	−37.40%	5035.45	−27.42%	3947.25	−40.44%	624.13	14,028.94	8054.16
SSP585 (2081–2100)	22,191.7	102.25%	5801.39	−33.82%	3856.25	−44.42%	1455.55	−78.04%	0	10,266.41	11,816.69

## Data Availability

The datasets supporting the conclusions of this article are included within the article. The datasets used and/or analyzed during the current study are available from the corresponding author upon reasonable request.

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
