# Peer review of "Comparative Study of Potential Habitats for Simulium qinghaiense (Diptera: Simuliidae) in the Huangshui River Basin, Qinghai–Tibet Plateau: An Analysis Using Four Ecological Niche Models and Optimized Approaches"

_insects, 2024, doi:10.3390/insects15020081_

Round 1
Reviewer 1 Report
Comments and Suggestions for Authors
The article by Liu et al. focuses on assessing suitable areas for Simulium qinghaiense in the Huangshui River Basin, Qinghai-Tibet Plateau, utilizing Four Ecological Niche Models and Optimized Approaches. This contribution presents valuable insights into the application of these approaches for predicting habitat suitability and insect distribution in the specified area. However, despite its intriguing content, the article contains some instances of inappropriate terminology, such as ‘potential suitable’ and ‘many simuliidae.’ Additionally, there is inconsistency in the use of the term "Simuliidae," the family name, throughout the text, leading to confusion. Hence, it is advisable for the authors to thoroughly review the manuscript to ensure consistent and accurate terminology usage. Furthermore, the text requires thorough proofreading for improved English clarity. The abstract is overly lengthy, and both the discussion and conclusion sections need enhancement. In my humble opinion, with due respect, the current version of the manuscript may not be ready for publication.

It is advisable for the authors to thoroughly review the manuscript to ensure consistent and accurate terminology usage. Furthermore, the text requires thorough proofreading for improved English clarity.
Author Response
Dear colleagues,
The manuscript insects-2817798 has been carefully revised. We appreciate the detailed and valuable comments and suggestions from you. This manuscript has been carefully revised according to your comments and suggestion. Our point-by-point responses to your comments are listed below.
Thanks again for your kind help!
With best wishes,
Hainan
Shao, Hainan
State Key Laboratory of Plateau Ecology and Agriculture
Qinghai University
Xining, Qinghai, 810016
China
Reviewer1:
Comparative Study of Potential Suitable Areas for Simulium qinghaiense (Diptera: Simuliidae) in the Huangshui River Basin, Qinghai-Tibet Plateau: An Analysis Using Four Ecological Niche Models and Optimized Approaches.
The article by Liu et al. focuses on assessing suitable areas for Simulium qinghaiense in the Huangshui River Basin, Qinghai-Tibet Plateau, utilizing Four Ecological Niche Models and Optimized Approaches. This contribution presents valuable insights into the application of these approaches for predicting habitat suitability and insect distribution in the specified area.
Re: Thanks a lot for your positive comments on our work.
However, despite its intriguing content, the article contains some instances of inappropriate terminology, such as ‘potential suitable’ and ‘many simuliidae.’ Additionally, there is inconsistency in the use of the term "Simuliidae," the family name, throughout the text, leading to confusion. Hence, it is advisable for the authors to thoroughly review the manuscript to ensure consistent and accurate terminology usage.
Re: Thank you for pointing out these mistakes. We have reviewed the manuscript thoroughly and carefully and the inappropriate usage of terminology has been revised.
Furthermore, the text requires thorough proofreading for improved English clarity.
Re: Thanks for your comment. We have been carefully revised manuscript and we have invited a native speaker to review our manuscript thoroughly.
The abstract is overly lengthy, and both the discussion and conclusion sections need enhancement.
Re: Thanks for your useful suggestion. The abstract, discussion and conclusion sections have been revised accordingly.
In my humble opinion, with due respect, the current version of the manuscript may not be ready for publication.
Re: Thank you for giving us a chance to revise the manuscript. We have revised the manuscript carefully according to your suggestions, we believe the current version will meet the required standards to be published in the journal.
Title
Kindly consider rephrasing the title by either eliminating the term “suitable” for enhanced clarity, or alternatively, revising it to better convey the intended meaning.
Re: Thank you for your comments and suggestions. We have changed the title of the manuscript to “Comparative Study of Potential Habitats for Simulium qinghaiense (Diptera: Simuliidae) in the Huangshui River Basin, Qinghai-Tibet Plateau: An Analysis Using Four Ecological Niche Models and Optimized Approaches”
Abstract
The abstract is excessively lengthy at approximately 430 words; it would be more effective to condense it to a concise 200-250 words for improved readability and clarity.
Re: Thanks for your suggestion. The abstract has been cut short according to your comments. (Line 43-80)
Line 17-18: “Despite its ecological importance, there is a current gap in research 17 regarding the potential suitable areas of this species within the region” better to say “Despite its ecological importance, there is currently a gap in research regarding the potential suitable areas of this species within the region.”
Re: Thanks for your comment. We have revised this sentence as follows: “Despite its ecological importance, there is currently a gap in research regarding the potential areas of this species within the region.”
Line 22: I recommend avoiding the use of both terms “potential suitable.” Authors are advised to choose either “potential” or “suitable” for clarity and precision.
Re: Thanks for your suggestion. We have used either “potential” or “suitable” in exactly the right place to express our statements clearly and precisely.
Line 53-54: (In the future, under various climate scenarios, it is anticipated that the suitable areas will diminish, signifying a contraction. The central distribution is projected to shift southeastward. Clarification is needed on the meaning of “contract” in this context.)
Re: Thanks for your comments. This sentence has been changed to “In the future, suitable areas under different climate scenarios are expected to contract, with the center of distribution shifting southeastward”. (Line 71)
Line 58: Simuliidae? Have the authors conducted a comprehensive survey covering all genera within this family? If not, it would be more informative to specify the particular species (as in article’s title) for which the study was conducted.
Re: Thanks for your comment. This research is mainly concentrated on one species of black flies, therefore we have replaced “Simuliidae” with “S. qinghaiense” as per your suggestion.
Introduction:
Line 103: please check the font size (adapt to)
Re: Thanks for your advice. We have revised the related text. (Line 108)
Line 118-120: Line 118-120: Please be mindful that ‘Simuliidae’ denotes a family name. Therefore, refrain from stating ‘many Simuliidae.’ Instead, you may choose to use terms such as ‘Simulids’ or‘species within the Simuliidae family’ for greater clarity. Kindly consider rephrasing the statement.
Re: Thanks for your comment. The inappropriate term “Simuliidae” has been replaced with “Simulids” as per your suggestion. (Line 128)
Line 96: Simuliidae (Diptera: Nematocera) are?
Re: Thank you for your question. “Simuliidae (Diptera: Nematocera)” has been changed to “Black flies (Diptera: Nematocera: Simuliidae)”. (Line 119)
Line 122-124: (In recent years, ecological niche models (ENMs) and species distribution models 122 (SDMs) have been widely applied to predict species responses to climate change, analyze 123 species distribution, and understand variations therein.) Please support this sentence with references.
Re: Thanks for your comment. We have added related references in the updated manuscript. (Line 160)
Lee-Yaw, J.A.; McCune, J.L.; Pironon, S.; Sheth, S.N. Species distribution models rarely predict the biology of real populations. Ecography 2022, e05877.
Materials and Methods
2.2. Software for Engineered Nanomaterials (ENMs) and Related Tools
In this subsection, I kindly urge the authors to enhance the clarity of my presentation by providing additional insights into the software utilized for handling Engineered Nanomaterials (ENMs) and associated tools. Requested details should encompass a comprehensive explanation of the software functionalities and operational mechanisms. Furthermore, I encourage authors to include direct links, especially if the software is accessible online for free, thereby facilitating readers’ ease of access and ensuring transparency in replicating the methodology.
Re: Thanks for your suggestion and comments. The detailed procedures of constructing the four models (e.g., Max Ent, GARP, BIOCLIM and DOMAIN models) were listed on section 2.5 (Line 281). Available online links have been added to the four models.
Discussion: In the discussion section, I suggest initiating with a concise summary of the key findings to provide readers with a quick overview. It is unnecessary to reiterate the methodology and present detailed results in the opening paragraph. The initial part of the current discussion appears to duplicate the methodology and abstract, which may impede the flow of the narrative. Streamlining this section to focus on the essential discoveries will enhance clarity and engagement for the reader. Subsequent paragraphs should be dedicated to delineating the outcomes of individual experiments, consolidating them into a singular comprehensive paragraph. Following this, engage in an interpretative analysis, aligning your findings with the extant body of research. This approach fosters clarity and avoids redundancy, enabling a more focused and impactful discussion.
Re: Thanks for your suggestions and comments. We have revised this section according to your comments in the updated manuscript.
Line 530: Pearson et al.’s study suggest that[59]?
Re: Thank you for pointing out the mistake. We have changed “Pearson et al.’s study suggest that” to “Previous studies suggested that”. (Line 581)
Conclusion:
This section serves the purpose of distilling key insights rather than delving into detailed findings. Its aim is to deliver a succinct and comprehensible message to the readers. Hence, I suggest a revision for clarity and brevity.
Re: Thanks very much for your valuable advice. We have deleted the duplicated contents and revised the conclusion for terse and concise. (Line 710)
Reviewer 2 Report
Comments and Suggestions for Authors
Overall, this was a well-written paper that covered a very interesting topic. In addition to the novel findings of model strength, it is particularly interesting how you described the range of the black fly species past, present and future.
The data used to construct the models was appropriate.
Below I have indicated some comments, minor corrections (for clarity and brevity), and raised a few basic questions for you to consider and address if you wish.
This is a really nice paper.
Simple summary
Page 1, lines 16-17. It would also seem that in addition to pathogen transmission, that the nuisance/annoyance impacts on livestock can be significant when black fly densities are high. Such annoyance can translate directly to economic losses for farmers.
Line 26: between ‘Quilian Mountains and provided’, I think you should insert a clarifier, perhaps ‘likely’ or ‘potentially’ because the model only predicts—you don’t know with absolute certainty the model prediction is true.
Lines 28-29: perhaps something like—“…as well as monitoring and early warning for threshold densities of S. quinghaiense. [I think it important to be specific on the species since not all black fly species are medically/economically important]
Abstract
Line 31: suggest inserting ‘black fly’ between endemic and species.
Pg. 2, line 56: “…River basin, but it also sheds light…”
Introduction
Page 2, line 83-84. Perhaps ‘A thorough [or full] comprehension…
Line 93. It’s not only quantity of insects but quality of insects (i.e., environmentally sensitive vs. tolerant). And I believe the quality is more important that abundance.
Page 3, Line 96. A suggestion: reference their common name here as well. Maybe ‘black flies (Diptera: Simuliidae) are….’ Then throughout the text use black flies occasionally instead. It makes for smoother reading than only using Simuliidae.
Lines 105-108. To simplify this section, I suggest you simply indicate that “…and distributed worldwide except Antarctica [reference numbers here].
Line 112. The sentence beginning “European Researchers…” should be started as a new paragraph.
Line 118: as far as black flies being vectors of human disease, I am only aware of onchocerciasis, which is known only from Africa and South America. So, S. quighaiense may be involved in the transmission of some zoonotic diseases in China (I don’t know what they are if so), but as far as humans and most livestock are concerned, they are really a nuisance, albeit a substantial one. I suggest you clarify this sentence to reflect that nuance.
Page 4, line 156. What exactly do you mean by ‘early warning’. Is it simply a warning that densities are getting high as an annoyance factor (since no human disease is involved, and what zoonotic diseases are of concern?)? Also, who is warned? Urban populations, farmers?
Line 170: you should identify the specific brand of GPS instrument and model number used to establish coordinates. Also, you really don’t need to use the word meticulously in line 1`71.
Page 6: environmental variables, starting line 208. It did not appear that current velocity nor discharge were selected as environmental variables for your analyses. But, black fly occurrences are strongly driven by faster currents. Why where those variables not included? Were they non-significant in the Pearson’s?
Also, some of your parameters in tables S.2 and S.3 may be autocorrelated since they are derived from the same data. Did you test for autocorrelation? If not, perhaps you should add language to indicate you accepted the risk, albeit low, of autocorrelation in your analysis.
Line 224. You need to provide a reference for the R package you used.
Line 233. Don’t start sentence with a number. Spell out 75% as Seventy-five percent.
Page 11, Figure 5. Just a comment that these figures are excellent for showing the scope of suitable areas for each model and how disparate they are.
Page 12, line 394: there is an extra s in areas.
Page 13, Table 4. The spacing on this table needs to be adjusted so the headers do not wrap. Perhaps check with the editor to see if you can use a slightly smaller font size.
Page 14, beginning line 429. This is a very interesting section presenting data for model projected suitable areas for S. quighaiense, but it doesn’t take into account water quality conditions (particularly degradation) that might occur during those same future time periods due to an expanding human population or other factors. While I do not expect you to include such unknowns in your model, I suggest that it might be worthwhile to present that possibility in the discussion.
General note: Check all of your figure and table legends. Some have sentence case (best choice) but others capitalize each word (poor choice). Each legend should have the same style.
Page 18, line 559. Your model is specific to only one species-- S. quighaiense. So, using the family name Simuliidae here is not appropriate since you did not evaluate data for other species—and not all species are described by the conditions you described (Cnephia in North America comes to mind). Use S. quighaiense here instead. You should do the same for the rest of this discussion section.
Page 19, starting line 624. Suggest instead of using the formal name Simuliidae in this instance use either black flies or simuliids. Doing so make it easier to read. Same in line 631. Also, in line 625, use ‘roles’ instead of ‘role’.
Line 631. Here again this references human disease, but I think the real issue for people will be annoyance from bites (possibly), and potential transmission of zoonotic disease to wildlife and livestock. I don’t recall any human diseases in the study area transmitted by black flies. If there are, list them specifically.
This paper has a simple summary, an abstract, and a conclusions section. They overlap extensively. Are all of these necessary? If this a journal requirement? If it is not, perhaps delete one or more of them with the preferred summary being the abstract.
Author Response
Dear colleague,
The manuscript insects-2817798 has been carefully revised. We appreciate the detailed and valuable comments and suggestions from you. This manuscript has been carefully revised according to your comments and suggestion. Our point-by-point responses to your comments are listed below.
Thanks again for your kind help!
With best wishes,
Hainan
Shao, Hainan
State Key Laboratory of Plateau Ecology and Agriculture
Qinghai University
Xining, Qinghai, 810016
China
Reviewer2:
Overall, this was a well-written paper that covered a very interesting topic. In addition to the novel findings of model strength, it is particularly interesting how you described the range of the black fly species past, present and future. The data used to construct the models was appropriate. Below I have indicated some comments, minor corrections (for clarity and brevity), and raised a few basic questions for you to consider and address if you wish.
Re: Thanks very much for your appreciation. We have revised this manuscript according to your comments and suggestions.
This is a really nice paper.
Simple summary
Page 1, lines 16-17. It would also seem that in addition to pathogen transmission, that the nuisance/annoyance impacts on livestock can be significant when black fly densities are high. Such annoyance can translate directly to economic losses for farmers.
Re: Thanks for your valuable comments. We have changed this sentence to “Notably, the female adults of S. qinghaiense could directly cause severe economic losses for farmers.” (Line 23)
Line 26: between ‘Quilian Mountains and provided’, I think you should insert a clarifier, perhaps ‘likely’ or ‘potentially’ because the model only predicts—you don’t know with absolute certainty the model prediction is true.
Re: Thanks for your comments. “a favorable refuge” has been changed to “a potentially favorable refuge”
Lines 28-29: perhaps something like—“…as well as monitoring and early warning for threshold densities of S. quinghaiense. [I think it important to be specific on the species since not all black fly species are medically/economically important]
Re: Thanks for your comments and suggestions. We have changed “Simuliidae” to “S. qinghaiense”.
Abstract
Line 31: suggest inserting ‘black fly’ between endemic and species.
Re: Thanks for your comment. We have made the modifications as per your request. (Line 44)
Pg. 2, line 56: “…River basin, but it also sheds light…”
Re: Thanks for your comment. We have deleted this sentence.
Introduction
Page 2, line 83-84. Perhaps ‘A thorough [or full] comprehension…
Re: Thanks for your comment. We have replaced “comprehensive” with “thorough” in the updated manuscript. (Line 105)
Line 93. It’s not only quantity of insects but quality of insects (i.e., environmentally sensitive vs. tolerant). And I believe the quality is more important that abundance.
Re: Thank you for your advice and comment. We have added “and quality (i.e., environmentally sensitive vs. tolerant)” to the updated manuscript. (Line 116)
Page 3, Line 96. A suggestion: reference their common name here as well. Maybe ‘black flies (Diptera: Simuliidae) are….’ Then throughout the text use black flies occasionally instead. It makes for smoother reading than only using Simuliidae.
Re: Thank you for your advice and comment. “Simuliidae (Diptera: Nematocera)” has been changed to “black flies (Diptera: Nematocera: Simuliidae)”. (Line 119)
Lines 105-108. To simplify this section, I suggest you simply indicate that “…and distributed worldwide except Antarctica [reference numbers here].
Re: Thanks for your suggestion. We have revised the manuscript accordingly. (Line 130)
Line 112. The sentence beginning “European Researchers…” should be started as a new paragraph.
Re: Thanks for your comment. We have made the modifications as per your request. (Line 139)
Line 118: as far as black flies being vectors of human disease, I am only aware of onchocerciasis, which is known only from Africa and South America. So, S. quighaiense may be involved in the transmission of some zoonotic diseases in China (I don’t know what they are if so), but as far as humans and most livestock are concerned, they are really a nuisance, albeit a substantial one. I suggest you clarify this sentence to reflect that nuance.
Re: Thanks for your suggestions. Related contents have been revised. (Line 145-157)
Black flies have gained notoriety due to the hematophagous tendencies exhibited by adult females across a multitude of species. Beyond merely posing a nuisance to humans, as well as domestic and wild animals, these flies are recognized as vectors for diseases, including bovine onchocerciasis and vesicular stomatitis virus in livestock. In tropical regions, anthropophilic species are involved in the transmission of mansonelliasis, a filarial infection, as well as onchocerciasis, colloquially known as 'river blindness'—ranking as the world’s second-leading infectious cause of blindness. Tokaoka (2012) has reported that the blackflies are parasite of wild animals implicated in the aetiology of seven zoonotic Onchocerca species in Japan. Although no studies documented the transmission of zoonotic diseases of S. qinghaiense to date in China, great attention should be paid to these black flies as potential vectors.
Here are some references:
Rivera, J.; Currie, D.C. Identification of nearctic black flies using DNA barcodes (Diptera: Simuliidae). Mol. Ecol. Resour. 2009, 9, 224–236.
Lourdes, E.Y.; Ya’cob, Z.; Low, V.L.; Izwan-Anas, N.; Mansor, M.S.; Dawood, M.M.; Takaoka, H.; Adler, P.H. Natural infections and distributions of parasitic Mermithids (Nematoda: Mermithidae) infecting larval black flies (Diptera: Simuliidae) in tropical streams of Malaysia. Acta. Trop. 2022, 230, 106386.
Pramual, P.; Adler, P.H. DNA barcoding of tropical black flies (Diptera: Simuliidae) of Thailand. Mol. Ecol. Resour. 2014, 14, 262–271.
Takaoka, H., Fukuda, M., Otsuka, Y., Aoki, C., Bain, S. Blackfly vectors of zoonotic onchocerciasis in Japan). Med. Vet. Entomol. 2012, 26, 372–378.
Page 4, line 156. What exactly do you mean by ‘early warning’. Is it simply a warning that densities are getting high as an annoyance factor (since no human disease is involved, and what zoonotic diseases are of concern?)? Also, who is warned? Urban populations, farmers?
Re: Thanks for your comment. Black flies are notorious for the haematophagous habits of the adult females of most species. Besides constituting a nuisance for humans and domestic and wild animals, black flies are known to be vectors of diseases such as bovine onchocerciasis and vesicular stomatitis virus in livestock. In tropical areas, anthropophilic species are implicated in the transmission of mansonelliasis (a filarial infection) and onchocerciasis or ‘river blindness’, the world’s second leading infectious cause of blindness. Tokaoka (2012) has reported that the blackflies are parasite of wild animals implicated in the aetiology of seven zoonotic Onchocerca species in Japan. In a few instances, saliva of some Simulium spp. has been associated with extensive tissue and organ pathology, including hemorrhagic shock and death. Although no studies documented the transmission of zoonotic diseases of S. qinghaiense to date in China, great attention should be paid to these black flies as potential vectors. In addition, the medical and socioeconomic impacts associated with black flies include reduced level of tourism and the death of domesticated birds and mammals. We think the monitoring and early warning of S. qinghaiense are very import to the whole distribution regions.
Related contents have been revised. (Line 199-200)
Line 170: you should identify the specific brand of GPS instrument and model number used to establish coordinates. Also, you really don’t need to use the word meticulously in line 1`71.
Re: Thanks for your advice. We have added the detailed brand and model of GPS instrument to the updated manuscript and deleted the word “meticulously”. (Line 214)
Page 6: environmental variables, starting line 208. It did not appear that current velocity nor discharge were selected as environmental variables for your analyses. But, black fly occurrences are strongly driven by faster currents. Why where those variables not included? Were they non-significant in the Pearson’s?
Re: Thanks for your comments and questions. Previous studies have indicated that the abundance of blackflies is higher and the young larvae exhibit faster growth rates in fast-flowing streams[Figueiró 2008; Brannin 2014]. It is worth noting the influence of current velocity on the S. qinghaiense. However, data on environmental factors are acquired from Worldclim, where the data on the current velocity of the Huangshui River Basin are lacking. Further study would take all the environmental factors into consideration. We have added the related content to the Disscussion. (Line 624-630)
Related references:
Figueiró, R., Nascimento, É.S.D., Gil-Azevedo, L., Maia-Herzog, M., Monteiro, R.F. Local distribution of blackfly (Diptera, Simuliidae) larvae in two adjacent streams: the role of water current velocity in the diversity of blackfly larvae. Rev. Bras. Entomol. 2008, 52, 452–454.
Brannin, M.T., O'donnell, M.K., Fingerut, J. Effects of larval size and hydrodynamicson the growth rates of the black fly Simulium tribulatum. Integr. Zool. 2014, 9, 61–69.
Also, some of your parameters in tables S.2 and S.3 may be autocorrelated since they are derived from the same data. Did you test for autocorrelation? If not, perhaps you should add language to indicate you accepted the risk, albeit low, of autocorrelation in your analysis.
Re: Thanks for your question. We have added the sentence “The potential issues of autocorrelation arising from environmental variables are negligible because the autocorrelation has low effects on correlation analysis.” to the manuscript. (line 260-261)
Line 224.You need to provide a reference for the R package you used.
Re: Thanks for your suggestion. We have added the reference “Cobos, M.E.; Peterson, A.T.; Barve, N.; Osorio-Olvera, L. Kuenm: An R package for detailed development of ecological niche models using maxent. PeerJ 2019, 7, e6281.” and an avaiable online link “https://github.com/marlonecobos/kuenm” to the updated manuscript. (line 276)
Line 233.Don’t start sentence with a number. Spell out 75% as Seventy-five percent.
Re: We are fairly sorry for the mistake. We have replaced “75%” with “Seventy-five percent”, other places with similar mistakes have also been revised. (Line 286, 293)
Page 11, Figure 5. Just a comment that these figures are excellent for showing the scope of suitable areas for each model and how disparate they are.
Re: Thanks very much for your appreciation.
Page 12, line 394: there is an extra s in areas.
Re: We are fairly sorry for the mistake. “areass” has been changed to “areas”.
Page 13, Table 4. The spacing on this table needs to be adjusted so the headers do not wrap. Perhaps check with the editor to see if you can use a slightly smaller font size.
Re: Thanks for your suggestion. After careful revision, this table and the headers were assigned on the same page.
Page 14, beginning line 429. This is a very interesting section presenting data for model projected suitable areas for S. qinghaiense, but it doesn’t take into account water quality conditions (particularly degradation) that might occur during those same future time periods due to an expanding human population or other factors. While I do not expect you to include such unknowns in your model, I suggest that it might be worthwhile to present that possibility in the discussion.
Re: Thanks for your comments and questions. We have added the content “In addition, significant alterations of local water quality caused by anthropogenic activities of humans such as expanding human population, intensive agricultural practices and releases from industrial wastewater and domestic sewage, greatly affect the distribution patterns of aquatic insects in the water, which might be another driving factor of distribution patterns of S. qinghaiense.” to the Discussion. (Line 666-671)
General note: Check all of your figure and table legends. Some have sentence case (best choice) but others capitalize each word (poor choice). Each legend should have the same style.
Re: We are fairly sorry for such mistakes. All figure and table legends have been checked and revised. (Line 784, 790)
Page 18, line 559. Your model is specific to only one species-- S. qinghaiense. So, using the family name Simuliidae here is not appropriate since you did not evaluate data for other species—and not all species are described by the conditions you described (Cnephia in North America comes to mind). Use S. qinghaiense here instead. You should do the same for the rest of this discussion section.
Re: Thanks for your comments and suggestions. We have replaced “Simuliidae” with “S. qinghaiense” in the updated manuscript.
Page 19, starting line 624. Suggest instead of using the formal name Simuliidae in this instance use either black flies or simuliids. Doing so make it easier to read. Same in line 631. Also, in line 625, use ‘roles’ instead of ‘role’.
Re: Thanks for your advice and comments. We have replaced “Simuliidae” with “black flies” and the “role” has been changed to “roles”. (Line 695)
Line 631. Here again this references human disease, but I think the real issue for people will be annoyance from bites (possibly), and potential transmission of zoonotic disease to wildlife and livestock. I don’t recall any human diseases in the study area transmitted by black flies. If there are, list them specifically.
Re: Thanks for your suggestions. Related contents have been revised. (Line 706-708)
Black flies are notorious for the haematophagous habits of the adult females of most species. Besides constituting a nuisance for humans and domestic and wild animals, black flies are known to be vectors of diseases such as bovine onchocerciasis and vesicular stomatitis virus in livestock. In tropical areas, anthropophilic species are implicated in the transmission of mansonelliasis (a filarial infection) and onchocerciasis or ‘river blindness’, the world’s second leading infectious cause of blindness. Tokaoka (2012) has reported that the blackflies are parasite of wild animals implicated in the aetiology of seven zoonotic Onchocerca species in Japan. Although no studies documented the transmission of zoonotic diseases of S. qinghaiense to date in China, great attention should be paid to these black flies as potential vectors.
The sentence “Currently, certain areas of the Huangshui River Basin may face increased risks related to disease transmission by black flies. Changes in the distribution center of this species may alter the regions at risk for disease, impacting the health of the surrounding human population. In summary, the results of this study are expected to provide a scientific basis for the prevention, control, and monitoring of zoonotic diseases spread by these insects.” has been changed to “The results of this study are expected to offering a scientific foundation for the prevention and control of the potential vectors of zoonotic diseases.”.
This paper has a simple summary, an abstract, and a conclusions section. They overlap extensively. Are all of these necessary? If this a journal requirement? If it is not, perhaps delete one or more of them with the preferred summary being the abstract.
Re: Thanks for your suggestions. According to the instructions for authors of this journal, these sections are all required. We have simplified the simple summary and abstract. Meanwhile, we have revised the conclusion in order to deliver a succinct and comprehensible message to the readers.
Reviewer 3 Report
Comments and Suggestions for Authors
The authors present a thorough analyses on potential spread and establishment of an insect species in China with implication for human and livestock health.
It is a very valuable work but it has to be presented in a succinct manner. It is very difficult to follow all the models and acronyms. The methods should be presented by model, followed by the charts and tables comparing the models in the results. It is difficult to follow all the models simultaneously being described.
The authors did follow this strategy for the section “2.4. Optimizing MaxEnt Model and Constructing the Four Models “ but then all the models are described simultaneously in the same sentences.
There is a section reporting 3. Results 3.1. Optimization Results of the MaxEnt Model
But all the other models are not listed and the rest of results are all models combined.
It is just difficult to follow.
The abstract is 434 words. It is too big.
“Precipitation of warmest quarter is conducive to survival in the range of 192–236 mm, with the optimum value at 236 mm; Altitude in the range of 2480–3192 m is favorable for survival, with the optimum value at 2480 m;” – Why? How was this conclusion made based on Figure 4 which is not based on any model? There are many instances in the text where sentences of this nature need to be explained if they are not based on the models, which seems to be the case for figure 4.
I don’t understand the biology and importance of the Diptera species as the authors list it as “They play a pivotal role in river ecosystems”, but at the same time say they are of concern as a vector for pathogens. This needs to be clear in the introduction.
Also, Based on Araújo's methodology[33] - there is no information on this methodology although it seems to be the base for all the work done in the Manuscript. The reader needs to know what Araujo did, without having to read Araujo work.
A one-way analysis of variance (ANOVA) was conducted to compare the AUC and 310 Kappa values among the four models -- Was normality and independence of variables attested for? An ANOVA required normal distribution of the data which being climatic data it is very uncommon.
Comments on the Quality of English LanguageMinor changes needed. For instance use of past tense or present tense verbs. Please choose only one.
Acquired data for 19 bioclimatic variables" - replace by Data was acquired...
Author Response
Dear colleague,
The manuscript insects-2817798 has been carefully revised. We appreciate the detailed and valuable comments and suggestions from you. This manuscript has been carefully revised according to your comments and suggestion. Our point-by-point responses to your comments are listed below.
Thanks again for your kind help!
With best wishes,
Hainan
Shao, Hainan
State Key Laboratory of Plateau Ecology and Agriculture
Qinghai University
Xining, Qinghai, 810016
China
Reviewer3:
The authors present a thorough analyses on potential spread and establishment of an insect species in China with implication for human and livestock health. It is a very valuable work but it has to be presented in a succinct manner. It is very difficult to follow all the models and acronyms. The methods should be presented by model, followed by the charts and tables comparing the models in the results. It is difficult to follow all the models simultaneously being described.
Re: Thanks very much for your appreciation. We have revised this manuscript according to your comments and suggestions.
The authors did follow this strategy for the section “2.4. Optimizing MaxEnt Model and Constructing the Four Models “ but then all the models are described simultaneously in the same sentences.
Re: Thanks for your suggestion. We have divided the previous 2.4 into two parts (i.e., 2.4 and 2.5) to avoid confusion for readers. (Line 268, 281)
There is a section reporting 3. Results 3.1. Optimization Results of the MaxEnt Model. But all the other models are not listed and the rest of results are all models combined. It is just difficult to follow.
Re: Thanks for your comment. In the Methods and Materials, we have divided previous section 2.4 into two parts (i.e., 2.4: Optimizing MaxEnt Model and 2.5: Constructing the Four Models: MaxEnt, GARP, BIOCLIM, and DOMAIN). After carefully conducting abundant literatures, only the R package for optimizing the MaxEnt model has been issued to date. Therefore, we just presented the optimization results of MaxEnt model in our study.
The abstract is 434 words. It is too big.
Re: Thanks for your comment. We have cut short the abstract according to your comments. (Line 43, 80)
“Precipitation of warmest quarter is conducive to survival in the range of 192–236 mm, with the optimum value at 236 mm; Altitude in the range of 2480–3192 m is favorable for survival, with the optimum value at 2480 m;” – Why? How was this conclusion made based on Figure 4 which is not based on any model? There are many instances in the text where sentences of this nature need to be explained if they are not based on the models, which seems to be the case for figure 4.
Re: Thanks for your comments and questions. The response curves of dominant environmental factors were simulated by MaxEnt model (Fig 4). Precipitation of warmest quarter is conducive to survival in the range of 192–280 mm, with the optimum value at 236 mm; Altitude in the range of 1,780–3,192 m is favorable for survival, with the optimum value at 2,480 m; Annual precipitation in the range of 348–475 mm is favorable for survival, with the optimum value at 412 mm; Precipitation of driest month is beneficial for survival in the range of -0.9–0.774 mm. These conclusions were calculated based on the response curves in the updated Fig 4. The relevant content “The jackknife test is employed to assess the importance of environmental variables and calculate the response curves of dominant environmental factors. This process is repeated ten times.” has been added to the materials and methods. (line 287-289)
I don’t understand the biology and importance of the Diptera species as the authors list it as “They play a pivotal role in river ecosystems”, but at the same time say they are of concern as a vector for pathogens. This needs to be clear in the introduction.
Re: Thanks for your suggestions. We have stated clearly the importance of the Diptera species in the introduction.
The black flies undergo complete metamorphosis, with larvae and pupae residing in water, feeding on organic particles in the water flow and attaching to various substrates. Their high population densities make them a key element in the energy transfer of river food chains, serving as a food source for many vertebrates and invertebrates. In addition, the immature stages of black flies typically require clean, unpolluted water. Significant alteration of local water quality could greatly affect the distribution patterns of S. qinghaiense in the water. Variability in the distribution and relative abundance of S. qinghaiense can act as an important indicator to evaluate water quality. Therefore, they play key roles in river ecosystems. Black flies are notorious for the haematophagous habits of the adult females of most species. Besides constituting a nuisance for humans and domestic and wild animals, black flies are known to be vectors of diseases such as bovine onchocerciasis and vesicular stomatitis virus in livestock. In tropical areas, anthropophilic species are implicated in the transmission of mansonelliasis (a filarial infection) and onchocerciasis or ‘river blindness’, the world’s second leading infectious cause of blindness. Tokaoka (2012) has reported that the blackflies are parasite of wild animals implicated in the aetiology of seven zoonotic Onchocerca species in Japan. Although no studies documented the transmission of zoonotic diseases of S. qinghaiense to date in China, great attention should be paid to these black flies as potential vectors of pathogens. (line 145-157)
Also, Based on Araújo's methodology - there is no information on this methodology although it seems to be the base for all the work done in the Manuscript. The reader needs to know what Araujo did, without having to read Araujo work.
Re: Thanks for your advice and comments. Araújo and New (2007) found that the projections by alternative models can be so variable as to compromise their usefulness for guiding policy decisions. However, significant improvements can be achieved if multiple models within an ensemble forecasting framework are used and the results are analyzed appropriately. Therefore, our study was designed on the basis of Araújo and New’ conclusion. We have changed “Based on Araújo's methodology [33], and incorporating field survey data, this study embraced the concept of an ensemble prediction system.” to “Given that projections by alternative models can deliver variable results [Araújo and New (2007)], this study embraced the concept of an ensemble prediction system incorporating field survey data.”. (Line 175-179)
A one-way analysis of variance (ANOVA) was conducted to compare the AUC and 310 Kappa values among the four models -- Was normality and independence of variables attested for? An ANOVA required normal distribution of the data which being climatic data it is very uncommon.
Re: Thanks for your suggestions. A Kolmogorov- Smirnov test was conducted to test the normality of the AUC and Kappa values of the four models (MaxEnt, GARP, BIOCLIM, and DOMAIN) (Table S4). Our results demonstrated that both the AUC and Kappa values of the four models exhibited a normal distribution (P>0.05), details are shown in Table S4. We have added table S4 to the supplement materials in the updated manuscript.
Comments on the Quality of English Language
Minor changes needed. For instance use of past tense or present tense verbs. Please choose only one.
Re: Thanks for your comment. We have thoroughly revised the manuscript in the updated manuscript according to your comment.
Acquired data for 19 bioclimatic variables"- replace by Data was acquired...
Re: Thanks for your comment. The sentence “Acquired data for 19 bioclimatic variables...” has been changed to “Data was acquired for 19 bioclimatic variables...”. (Line 231)
Round 2
Reviewer 1 Report
Comments and Suggestions for Authors
The authors have incorporated the suggestions provided, which enhances the quality of the manuscript. Therefore, the manuscript can be considered for publication. One minor suggestion is that the species name 'Simulium qinghaiense' in the article title should be italicized, in accordance with proper scientific writing conventions.